# Robust Consensus Anchor Learning for Efficient Multi-view Subspace Clustering

**Yalan Qin** [1]  **Nan Pu** [2]  **Guorui Feng** [1]  **Nicu Sebe** [2]

## Abstract

As a leading unsupervised classification algorithm in artificial intelligence, multi-view subspace clustering segments unlabeled data from different subspaces. Recent works based on the anchor have been proposed to decrease the computation complexity for the datasets with large scales in multi-view clustering. The major differences among these methods lie on the objective functions they define. Despite considerable success, these works pay few attention to guaranting the robustness of learned consensus anchors via effective manner for efficient multi-view clustering and investigating the specific local distribution of cluster in the affine subspace. Besides, the robust consensus anchors as well as the common cluster structure shared by different views are not able to be simultaneously learned. In this paper, we propose Robust Consensus anchors learning for efficient multi-view Subspace Clustering (RCSC). We first show that if the data are sufficiently sampled from independent subspaces, and the objective function meets some conditions, the achieved anchor graph has the block-diagonal structure. As a special case, we provide a model based on Frobenius norm, non-negative and affine constraints in consensus anchors learning, which guarantees the robustness of learned consensus anchors for efficient multi-view clustering and investigates the specific local distribution of cluster in the affine subspace. Experiments performed on eight multi-view datasets confirm the superiority of RCSC based on the effectiveness and efficiency.

---

[1] School of Communication and Information Engineering, Shanghai University, Shanghai 200444, China [2]Department of Information Engineering and Computer Science, University of Trento, Trento 40128, Italy. Correspondence to: Guorui Feng <fgr2082@aliyun.com>.

*Proceedings of the 42nd International Conference on Machine Learning*, Vancouver, Canada. PMLR 267, 2025. Copyright 2025 by the author(s).

## 1. Introduction

Clustering is an important field in artificial intelligence and machine learning (Jain, 2008). As the information technology develops, large amounts of data from multiple views or channels (Liu et al., 2025; Qin et al., 2023d) can be collected in real-world scenarios. This type of data is termed as multi-view data and it widely exists in the world. For instance, we can depict an image by multiple representations, such as local binary pattern (LBP), histogram of oriented gradient (HOG) and Gabor feature representation. For dealing with the multi-view data in clustering tasks, various multi-view clustering approaches have been presented in the literature. As opposed to clustering for the data with single view (Qin et al., 2023e;a; 2021; 2022b; 2025b; Pu et al., 2024), multi-view clustering can achieve more reasonable performance in practice (Qin et al., 2022a; Liu et al., 2024; Qin et al., 2023b; Liu et al., 2023b; Qin et al., 2023c; Sun et al., 2024; Qin et al., 2024a; Liu et al., 2022a; Qin et al., 2024d; Liu et al., 2023a; Qin et al., 2025c; Liu et al., 2022b; Qin et al., 2025e). It is able to integrate diverse feature representations of an object obtained by different views and provide more comprehensive object information. Among the existing multi-view clustering works, methods based on the graph have gained significant attention in recent years.

The graph-based methods for multi-view clustering (Nie et al., 2017b; Zhan et al., 2019; Wang et al., 2020) can reflect the relationships among samples in multi-view data by constructing graph structures. Then the final results can be achieved based on the partiton of the obtained graph. These methods usually adopt an $n \times n$ adjacent graph to formulate the relationships among data points. Cao et al. (Cao et al., 2015) presented a multi-view clustering method based on diversity and smoothness, which investigates the complementarity among different representations. Wang et al. (Wang et al., 2020) learned the unified global graph and view-specific graphs by adopting the mutual reinforcement technique. Chen et al. (Chen et al., 2020) simultaneously learned the global self-representation, latent embedding space and cluster structure for subspace learning on multi-view data. Liang et al. (Liang et al., 2020) utilized the multi-view graph learning to learn a unified graph and

leveraged the incosistency and consistency among multiple view-specific graphs. Nie et al. (Nie et al., 2017a) presented to simultaneously perform local structure learning and semi-supervised classification/clustering. Wang et al. (Wang et al., 2020) coupled unified graph, graph induced by similarity across views and indicator into a unified framework. Despite significant progress, most of these multi-view clustering approaches tend to suffer from high computation complexity. It takes $O(n^2)$ to construct an $n \times n$ adjacent graph and needs $O(n^3)$ in partitioning this graph, which limits their scalability for multi-view datasets with large scales.

For dealing with the computation complexity issue, multi-view clustering methods based on the anchor have been given (Qin et al., 2025a;d; Qin & Qian, 2024; Qin et al., 2024c;b), which show promising capability in real applications. Different from constructing an $n \times n$ graph, the methods based on the anchor typically produce small number of anchors from the dataset and represent the structure of data by building an $n \times l$ anchor graph, where $l$ denotes the total number of anchors. In general, $l$ is lower than $n$, which enables the scale of data to be greatly decreased for reaching the goal of increasing the clustering efficiency. Specifically, Yang et al. (Yang et al., 2021) increased the efficiency and all views are required to yield the same anchors. Kang et al. (Kang et al., 2020) employed the subspace learning based on the anchor to learn a anchor graph for each view and then heuristically combined different anchor graphs into a unified one. Li et al. (Li et al., 2022) presented a scalable multi-view clustering method by fusing anchor graphs. It is able to employ the discrete cluster structure to adaptively learn a unified graph by anchor graph fusion. Wang et al. (Wang et al., 2022) employed multiple projection matrices and a set of latent consensus anchors to learn a unified anchor graph. Yang et al. (Yang et al., 2022) seeked for $l$ anchors on the original data with the guidance of $K$-means. The produced centroids are regarded as anchors and a sparse anchor graph is constructed between the obtained anchors and the original dataset. Despite great success, these methods pay few attention to ensuring the robustness of learned consensus anchors for efficient multi-view clustering and investigating the specific local distribution of cluster in the affine subspace. The correlation among the learned consensus anchors, which encourages the grouping effect and tends to group highly correlated anchors together, is not able to be fully explored. Besides, the robust consensus anchors and the common cluster structure shared by different views are not able to be simultaneously learned in a unified framework. Then the mutual enhancement for these procedures is not guaranteed in this manner and more discriminative consensus anchors as well as cluster indicator are not obtained.

To cope with the above issues, we propose a novel Robust Consensus anchor learning for efficient multi-view Subspace Clustering (RCSC). We first theoretically show that a block-diagonal anchor graph can be obtained if the objective function meets certain conditions based on the independent subspace assumption. As a special case, we provide a model based on Frobenius norm, non-negative and affine constraints in consensus anchors learning, which guarantees the robustness of learned consensus anchors for efficient multi-view clustering and investigates the specific local distribution of cluster in the affine subspace. While it is simple, we theoretically give the geometric analysis regarding the formulated RCSC. The union of these three constraints is able to restrict how each data point is described in the affine subspace with specific local distribution of cluster for guaranting the robustness of learned consensus anchors. We can fully explore the correlation among the learned consensus anchors with the view-specific projection, which encourages the grouping effect and groups highly correlated anchors together. The robust anchor learning, partition and anchor graph construction are jointly modeled in a unified framework. Then the robust consensus anchors and the common cluster structure shared by different views are able to be simultaneously learned. We can guarantee the mutual enhancement for these procedures in this manner and achieve more discriminative consensus anchors as well as the cluster indicator. By imposing the orthogonal constraints on the actual bases, we constrain a factor matrix to be the cluster indicator matrix based on the rigorous clustering interpretation. We then develop an alternate minimizing algorithm for solving the formulated problem. The major contributions in this paper are:

1. We propose a novel Robust Consensus anchor learning for efficient multi-view Subspace Clustering (RCSC). We first theoretically demonstrate that an anchor graph with block-diagonal structure can be achieved if the objective function satisfies certain conditions. As a special case, we give a model based on Frobenius norm, non-negative and affine constraints in consensus anchors learning, which guarantees the robustness of learned consensus anchors for efficient multi-view clustering and investigates the specific local distribution of cluster in the affine subspace.

2. We are able to fully explore the correlation among the learned consensus anchors with the guidance of view-specific projection, which encourages the grouping effect and tends to group highly correlated anchors together. We jointly perform the robust anchor learning, partition and anchor graph construction in a unified framework. Then, the robust consensus anchors and the common cluster structure shared by different views are simultaneously learned, which ensures the

mutual enhancement for these procedures and helps lead to more discriminative consensus anchors as well as the cluster indicator.

3. We impose the orthogonal constraints on the actual bases and constrain a factor matrix to be the cluster indicator matrix built on the rigorous clustering interpretation. Extentive experiments on different multi-view datasets validate the effectivenss and efficiency of RCSC, especially on the datasets with large scales.

## 2. Methodology

In this section, we present the motivation and formulation of RCSC, followed by the optimization process and the related analysis of computation complexity for RCSC.

### 2.1. Motivation

The anchor strategy is usually employed to find the underlying structure by choosing a small number of data points as anchor bases. Some existing mutli-view clustering methods based on the anchor conduct $K$-means to achieve clustering centroids with the anchor bases being fixed. Despite great success, these methods pay few attention to guaranting the robustness of learned consensus anchors for efficient multi-view clustering and investigating the specific local distribution of cluster in the affine subspace. The correlation among the learned consensus anchors, which encourages the grouping effect and groups highly correlated anchors together, is ignored to be fully explored. Moreover, the robust consensus anchors and the common cluster structure shared by multiple views are not simultaneously learned. Therefore, the mutual enhancement for these procedures is not effectively ensured and more discriminative consensus anchors as well as cluster indicator are not acquired.

### 2.2. Formulation

We generate view-specific data points via certain generation model based on a latent space. Given multi-view dataset $\{X^p \in R^{d_p \times n}\}_{p=1}^v$ with $d_p$ and $n$ being the dimension and size of dataset, we first assume that the data are noise free and formulate the corresponding objective function as:

$$\min_{U^p, A, S} \sum_{p=1}^v ||X^p - U^p A S||_F^2, \qquad (1)$$
$$s.t. \ (U^p)^T U^p = I, \ A^T A = I,$$

where $S \in R^{l \times n}$ is the shared affinity matrix, $\{U^p\}_{p=1}^v \in R^{d_p \times d}$ indicates a projection matrix as the consensus anchor guidance, $A \in R^{d \times l}$ represents the unified anchors, $l$ and $d$ are the number of anchors and shared dimension

across views, respectively. $U^p A \in R^{d_p \times l}$ represents the basis matrix. We then theoretically demonstrate that a block-diagonal anchor graph can be achieved if the corresponding objective function satisifies certain conditions based on the independent subspace assumption, which is shown as Theorem 1 in the following.

**Theorem 1.** *Assume the subspaces* $\{\Omega_i\}_{i=1}^k$ *are independent,* $X_i^p$ *is a matrix with columns consisting of some vectors from* $\Omega_i$ *and* $U_i^p A_i$ *is a matrix with columns consisting of a basis of* $\Omega_i$, *where* $(U^p)^T U^p = I$ *and* $A^T A = I$. *The solution S to the following form*

$$X^p = (U^p A) S \qquad (2)$$

*is block-diagonal and unique.*

**Proof.** For data point $x^p \neq 0$ and $x^p \in \Omega_i$, we just need to prove that there exists a unique $s$, $x^p = (U^p A)s$, where $s = [s_1^T, \cdots, s_k^T]$, with $s_i \neq 0$ and $s_j = 0$ for all $j \neq i$. Since $(U^p)^T U^p = I$ and $A^T A = I$, we can obtain $(U^p A)^T (U^p A) = A^T (U^p)^T U^p A = I$. Thus, $U^p A$ is orthogonal. Due to the orthogonality among subspaces, there exists a unique decomposition for $x^p$ as follows:

$$x^p = 0 + \cdots + x^p + \cdots + 0$$
$$= (U_1^p A_1) s_1 + \cdots + (U_i^p A_i) s_i + \cdots + (U_k^p A_k) s_k, \qquad (3)$$

where $(U_i^p A_i) s_i \in \Omega_i$ and $i = 1, \cdots, k$. Therefore, $(U_i^p A_i) s_i = x^p$ and $(U_j^p A_j) s_j = 0$ for all $j \neq i$. Since $U_i^p A_i$ is a basis of $\Omega_i$, we have $s_i \neq 0$, $s_i$ is unique, and $s_j = 0$ for all $j \neq i$.

Accoring to Theorem 1, a basis of $X^p$ can be learned and we use it as the dictionary. It is easy to obtain a true solution by solving the problem in Eq. (1) if subspaces are independent. To deal with more general multi-view clustering issue in the affine subspace, we introduce the affine constraint $S^T \mathbf{1} = 1$ to the original objective function and obtain the optimization problem as:

$$\min_{U^p, A, S} \sum_{p=1}^v ||X^p - U^p A S||_F^2, \ \ s.t. \ S^T \mathbf{1} = 1, \qquad (4)$$
$$(U^p)^T U^p = I, \ A^T A = I,$$

where $\mathbf{1}$ denotes the vector with all entries being one. To locally depict the distribution of contaminated data points from different subspaces, we add a non-negative constraint for $S$ and achieve the problem as:

$$\min_{U^p, A, S} \sum_{p=1}^v ||X^p - U^p A S||_F^2, \ \ s.t. \ S \geq 0, \ S^T \mathbf{1} = 1, \qquad (5)$$
$$(U^p)^T U^p = I, \ A^T A = I.$$

Thus, the non-negative constraint endows the learned $S$ with the probabilistic meaning. In real applications, the data are usually contaminated with the possible noise. We then adopt the Frobenius norm for penalizing the noise based on affine and non-negative constraint in Eq. (5), formulated as:

$$\min_{U^p,A,S} \sum_{p=1}^{v} ||X^p - U^p AS||_F^2 + \lambda||S||_F^2, \ s.t. \ S \geq 0, \tag{6}$$
$$S^T \mathbf{1} = 1, \ (U^p)^T U^p = I, \ A^T A = I,$$

where $\lambda > 0$ is a parameter to balance different parts. We then guarantee the robustness for efficient multi-view clustering and investigate the specific local distribution of cluster in the affine subspace. The grouping effect is stated in the theorem as follows.

**Theorem 2.** *Given data point $x^p \in R^{d_p}$, matrix $U^p A \in R^{d_p \times n}$ and parameter $\lambda$. Assume each column of $U^p A$ is normalized. Let $s^*$ be the optimal solution to the following problem:*

$$\min_{U^p,A,s} \sum_{p=1}^{v} ||x^p - U^p As||_2^2 + \lambda||s||_2^2, \ \ s.t. \ S \geq 0, \tag{7}$$
$$S^T \mathbf{1} = 1, \ (U^p)^T U^p = I, \ A^T A = I.$$

*We have*

$$\frac{||s_i^* - s_j^*||_2}{||x^p||_2} \leq \frac{1}{\lambda}\sqrt{2(1-r)}, \tag{8}$$

*where $r = (u_i^p a_i)^T (u_j^p a_j)$ denotes the basis correlation.*

**Proof.** Let $L(s) = ||x^p - U^p As||_2^2 + \lambda||s||_2^2$. Since $s^*$ is the optimal solution to Eq. (7), it meets

$$\left. \frac{\partial L(s)}{\partial s_k} \right|_{s=s^*} = 0. \tag{9}$$

Since $(U^p)^T U^p = I$ and $A^T A = I$, we can obtain $(U^p A)^T (U^p A) = A^T (U^p)^T U^p A = I$. Thus, $U^p A$ is orthogonal. Then we have

$$-2(u_i^p a_i)^T (x^p - U^p As^*) + 2\lambda s_i^* = 0, \tag{10}$$

$$-2(u_j^p a_j)^T (x^p - U^p As^*) + 2\lambda s_j^* = 0, \tag{11}$$

Thus

$$s_i^* - s_j^* = \frac{1}{\lambda}((u_i^p a_i)^T - (u_j^p a_j)^T)(x^p - U^p As^*), \tag{12}$$

$$||u_i^p a_i||^2 = ||u_j^p a_j||^2 = 1, \tag{13}$$

and

$$(u_i^p a_i)^T u_j^p a_j = 0. \tag{14}$$

Then we have $||u_i^p a_i - u_j^p a_j||_2 = \sqrt{2}$. Note that $s^*$ is the optimal solution to Eq. (7), we can obtain

$$||x^p - U^p As^*||_2^2 + \lambda||s^*||_2^2 = L(s^*) \leq L(0) = ||x^p||_2^2. \tag{15}$$

Therefore, $||x^p - U^p As^*||_2 \leq ||x^p||_2$. Thus, Eq. (15) implies

$$\frac{||s_i^* - s_j^*||_2}{||x^p||_2} \leq \frac{\sqrt{2}}{\lambda}. \tag{16}$$

The grouping effect in the above theorem demonstrates that the obtained solution is correlation dependent. Theorem 2 shows that the difference between $s_i^*$ and $s_j^*$ is nearly zero if $u_i^p a_i$ and $u_j^p a_j$ are highly correlated. We then fully explore the correlation among the learned consensus anchors $A$ with the guidance of view-specific projection $U^p$, which encourages the grouping effect and groups highly correlated anchors together.

As abovementioned, we introduce the Frobenius norm, non-negative and affine constraints into the shared affinity matrix learning. While it is simple, the union of these three constraints can restrict how each data point is described in the affine subspace with specific local distribution of cluster for guaranting the robustness of learned consensus anchors. We then denote the basis matrix $U^p A$ as

$$U^p A = [(U^p A)^1, \cdots, (U^p A)^t, \cdots, (U^p A)^k], \tag{17}$$

where $(U^p A)^t$ is the basis matrix lying in the $t$-th affine subspace. We adopt $(U^p A)_{-i}^t$ to denote basis matrices in the $t$-th affine subspace except the basis $(U^p A)_i$. $(U^p A)^{-t}$ is employed to indicate basis matrices in all affine subspaces except the $t$-th affine subspace. The sets of basis in $(U^p A)^t$, $(U^p A)^{-t}$ and $(U^p A)_{-i}^t$ can be denoted by $\Gamma((U^p A)^t)$, $\Gamma((U^p A)^{-t})$ and $\Gamma((U^p A)_{-i}^t)$, respectively. A basis tends to locate in three possible positions regarding $(U^p A)^t$, i.e., edge, inside and vertex. We adopt $edge((U^p A)^t)$, $inside((U^p A)^t)$ and $vertex((U^p A)^t)$ to indicate these three positions. For non-vertex (edge and inside) basis, we first give the lemma and then have the theorem as follows:

**Lemma 1.** *Assume that $A$ and $D$ are square matrices, we can obtain*

$$\left|\left| \begin{pmatrix} A & B \\ C & D \end{pmatrix} \right|\right|_* \geq \left|\left| \begin{pmatrix} A & \mathbf{0} \\ \mathbf{0} & D \end{pmatrix} \right|\right|_* = ||A||_* + ||D||_* \ for$$

*matrices $B$ and $C$ with compatible dimension.*

**Theorem 3.** *For any non-vertex basis $(U^p A)_i \in edge((U^p A)^t)$ and $(U^p A)_i \in inside((U^p A)^t)$ with $t = 1, 2, \cdots, k$, the optimal solution to Eq. (6) is block diagonal if $\Gamma((U^p A)^t)$ does not intersect with $\Gamma((U^p A)^{-t})$.*

**Proof.** We prove the above theorem by the contrapositive and assume that there is an optimal solution $S$ to Eq. (6)

and $S$ does not have the block diagonal property. We set the block diagonal matrix $W$ as

$$W_{ij} = \begin{cases} S_{ij}, & \text{if } (U^p A)_i \text{ and } (U^p A)_j \text{ in same subspace,} \\ 0, & \text{otherwise.} \end{cases}$$
(18)

Then $Q = S - W$ is adopted to denote the difference between $S$ and $W$. We employ $\mathbb{C} = \{j : (U^p A)_j \in \Gamma((U^p A)^t)\}$ and $\Bbbk = \{j : (U^p A)_j \notin \Gamma((U^p A)^t)\}$ to indicate indices of basis for the $t$-th affine subspace and other affine subspaces, respectively. Each edge and inside basis $(U^p A)_i$ lying in $\Gamma((U^p A)^t)$ which causes $S$ against the block diagonal structure can be written as

$$\sum_{j \in \Bbbk} Q_{ij}(U^p A)_j = (U^p A)_i - \sum_{j \in \mathbb{C}} W_{ij}(U^p A)_j. \quad (19)$$

Since $\sum_{j \in \mathbb{C}} W_{ij} + \sum_{j \in \Bbbk} Q_{ij} = 1$, we divide two sides of Eq. (19) by $\sum_{j \in \Bbbk} Q_{ij}$ and obtain:

$$\frac{\sum_{j \in \Bbbk} Q_{ij}(U^p A)_j}{\sum_{j \in \Bbbk} Q_{ij}} = \frac{(U^p A)_i - \sum_{j \in \mathbb{C}} W_{ij}(U^p A)_j}{\sum_{j \in \Bbbk} Q_{ij}}$$
$$= \frac{(U^p A)_i - \sum_{j \in \mathbb{C}} W_{ij}(U^p A)_j}{1 - \sum_{j \in \mathbb{C}} W_{ij}}.$$
(20)

We observe that the left and right sides of Eq. (20) are basis in $\Gamma((U^p A)^{-t})$ and $\Gamma((U^p A)^t)$, respectively. It can be concluded that $\Gamma((U^p A)^{-t})$ does intersect with $\Gamma((U^p A)^t)$. Therefore, according to Lemma1, we have $\|S\|_* \geq \|W\|_*$. Then, $W$ is the optimal and owns the block diagonal structure. Though it is hard to give sufficient conditons of the similar structure for vertices as Theorem 3, we find that the proportions of vertices are far more less than edge and inside basis in practice.

To integrate the partition into the unified framework, we adopt the orthogonal and nonnegative factorization to directly assign clusters to the data. Then extra postprocessing steps are not needed in recovering cluster structures based on the factor matrix. Specifically, we impose the orthogonal constraint on the actual bases. The above process is formulated as:

$$\min_{\alpha, U^p, A, S, G, F} \sum_{p=1}^{v} \alpha_p^2 \|X^p - U^p A S\|_F^2 + \lambda \|S\|_F^2$$
$$+ \beta \|S - GF\|_F^2, \ s.t. \ S \geq 0, \ S^T \mathbf{1} = 1,$$
$$\alpha^T \mathbf{1} = 1, \ (U^p)^T U^p = I, \ A^T A = I, \ G^T G = I,$$
$$F_{ij} \in \{0, 1\}, \ \sum_{i=1}^{k} F_{ij} = 1, \ \forall j = 1, 2, \cdots, n,$$
(21)

where $\alpha_p^2$ denotes the learned coefficients, $\beta > 0$ is a parameter for balancing different terms, $G \in R^{l \times k}$ stands for

the centroid matrix and $F \in R^{k \times n}$ represents the cluster assignment with $F_{ij} = 0$ if $j$-th data point is not belonged to the $i$-th cluster and 1 otherwise.

## 2.3. Optimization

For solving the problem in Eq. (21), we design an alternate optimization algorithm to seek for the solution to each variable while fixing the other variables.

$U^p$**-subproblem**: With the other variables being fixed, the objective function regarding $U^p$ is

$$\min_{U^p} \sum_{p=1}^{v} \alpha_p^2 \|X^p - U^p A S\|_F^2, \ s.t. \ (U^p)^T U^p = I.$$
(22)

We then transform the above optimization problem by trace as follows:

$$\max_{U^p} Tr((U^p)^T C^p), \ s.t. \ (U^p)^T U^p = I, \quad (23)$$

where $C^v = X^p S^T A^T$. Assuming the singular value decomposition (SVD) of $C^v$ is $U_C' \Sigma_C V_C^T$, we can easily obtain the optimal $U^p$ by calculating $U_C' V_C^T$.

$S$**-subproblem**: With the other variables being fixed, the objective function regarding $S$ is

$$\min_{S} \sum_{p=1}^{v} \alpha_p^2 \|X^p - U^p A S\|_F^2 + \lambda \|S\|_F^2 + \beta \|S - GF\|_F^2,$$
$$s.t. \ S \geq 0, \ S^T \mathbf{1} = 1.$$
(24)

We then rewrite it by the quadratic programming (QP) problem as follows:

$$\min h^T S_{:,j} + \frac{1}{2} S_{:,j}^T W S_{:,j}, \ s.t. \ S \geq 0, \ S_{:,j}^T \mathbf{1} = 1, \quad (25)$$

where $h^T = -2 \sum_{p=1}^{v} (X_{:,j}^p)^T U^p A - 2\beta F_{:,j}^T G^T$ and $W = 2(\sum_{p=1}^{v} \alpha_p^2 + \lambda + \beta)I$. Thus, we tackle the QP problem to achieve the optimization for each column in $S$.

$A$**-subproblem**: With the other variables being fixed, the objective function regarding $A$ is

$$\min_{A} \sum_{p=1}^{v} \alpha_p^2 \|X^p - U^p A S\|_F^2, \ s.t. \ A^T A = I. \quad (26)$$

Likewise, Eq. (26) is equal to the problem as follows:

$$\max_{A} Tr(A^T B), \ s.t. \ A^T A = I, \quad (27)$$

where $B = \sum_{p=1}^{v} \alpha_p^2 (U^p)^T X^p S^T$. Then the optimal $A$ is $U_B' V_B^T$, where $B = U_B' \Sigma_B V_B^T$.

$F$-**subproblem**: With the other variables being fixed, the objective function regarding $F$ is

$$\min_F \beta \|S - GF\|_F^2, \quad s.t. \ F_{ij} \in \{0,1\},$$
$$\sum_{i=1}^{k} F_{ij} = 1, \ \forall j = 1, 2, \cdots, n. \tag{28}$$

We then independently solve each object for the optimization problem and obtain

$$\min_{F_{:,j}} \beta \|S_{:,j} - GF_{:,j}\|^2, \quad s.t. \ F_{:,j} \in \{0,1\}^k, \ \|F_{:,j}\|_1 = 1. \tag{29}$$

We can find the optimal row by

$$i^* = \arg\min_i \|S_{:,j} - G_{:,i}\|^2. \tag{30}$$

$G$-**subproblem**: With the other variables being fixed, the objective function regarding $G$ is

$$\min_G \beta \|S - GF\|_F^2, \quad s.t. \ G^T G = I. \tag{31}$$

Then the optimization problem regarding $G$ is rewritten as

$$\max_G Tr(G^T J), \quad s.t. \ G^T G = I, \tag{32}$$

where $J = SF^T$. Then, the optimal $G$ is equal to $U_J' V_J^T$, where $J = U_J' \Sigma_J V_J^T$.

$\alpha_p$-**subproblem**: With the other variables being fixed, the objective function regarding $\alpha_p$ is

$$\min_\alpha \sum_{p=1}^{v} \alpha_p^2 \|X^p - U^p AS\|_F^2, \quad s.t. \ \alpha^T \mathbf{1} = 1. \tag{33}$$

Based on Cauchy-Buniakowsky-Schwarz inequality, the optimal $\alpha_p$ can be obtained by

$$\alpha_p = \frac{\frac{1}{\|X^p - U^p AS\|_F}}{\sum_{p=1}^{v} \frac{1}{\|X^p - U^p AS\|_F}}. \tag{34}$$

The objective function monotonically decreases in each iteration until convergence since the convex property for each sub-problem. We list the procedure of RCSC in Algorithm 1.

## 2.4. Complexity Analysis

The computation cost of our method includes the costs brought by optimizing all variables. Specificlly, it costs $O(l^3 n)$ to update $S$. In optimizing $U^p$, conducting matrix multiplication needs $O(d_p d k^2)$ and SVD consumes $O(d_p d^2)$ for each view. It needs $O(dlk^2)$ in matrix multiplication and $O(dl^2)$ in SVD for optimizing $A$. The

---

**Algorithm 1** Algorithm of RCSC

**Input:** Multi-view dataset $\{X^p\}_{p=1}^v$, parameter $\lambda$, $\beta$, number of clusters $k$.
**Output:** Cluster assignment $F$.
1: **Initialize:** Initialize $A$, $U^p$, $\{\alpha_p\}_{p=1}^v$, $S$, $F$ and $G$.
2: **repeat**
3:      Update $S$ by solving Eq. (24);
4:      Update $\{U_p\}_{p=1}^v$ by solving Eq. (22);
5:      Update $A$ by solving Eq. (26);
6:      Update $G$ by solving Eq. (31);
7:      Update $F$ by solving Eq. (28);
8:      Update $\alpha$ by solving Eq. (33);
9: **until** convergence

---

complexity of $O(lnk)$ is needed to optimize $F$. It takes $O(lk^2 + lk^3)$ to update $G$, which consists of the time cost in SVD and matrix multiplication. It needs $O(1)$ to update $\alpha_p$. The total time cost of our method is $O((pd^2 + pdk^2 + dlk^2 + nl^3 + lk^2 + lk^3 + dl^2 + lnk)o)$ with $o$ being the total number of iterations, where $p = \sum_{p=1}^{v} d_p$. Since $n \gg k$ and $n \gg l$, the computation cost of our method is nearly linear to $O(n)$.

## 3. Experiments

In this part, we evaluate the proposed method against the representative methods on eight multi-view datasets under different metrics in terms of effectiveness and efficiency.

### 3.1. Datasets and Experimental Settings

For the experimental evaluation, we use eight real-world multi-view datasets, namely, ORL, Mfeat, Caltech101-20, Caltech101-all, SUNRGBD (Song et al., 2015), NUSWIDEOBJ (Chua et al., 2009), AWA and Youtube-Face. Eight representative multi-view clustering methods are employed for comparison, including AMGL (Nie et al., 2016), SFMC (Li et al., 2022), BMVC (Zhang et al., 2019), LMVSC (Kang et al., 2020), MSGL (Kang et al., 2022), FRMVS (Wang et al., 2022), EOMSC-CA (Liu et al., 2022c) and OMSC (Chen et al., 2022).

We need to determine the anchor number in evaluating the clustering performance of all methods. For ensuring the fairness, the best parameters are used for compared methods. The anchor number of our method is tuned in the range of $[2k, 3k, \cdots, 7k]$, where $k$ denotes the total number of clusters in dataset. To reduce the randomness, we repeat each experiment for 20 times and report their mean values and variances in the experiment. We evaluate the clustering results by three widely adopted metrics, which consists of accuracy (ACC), normalized mutual information (NMI)

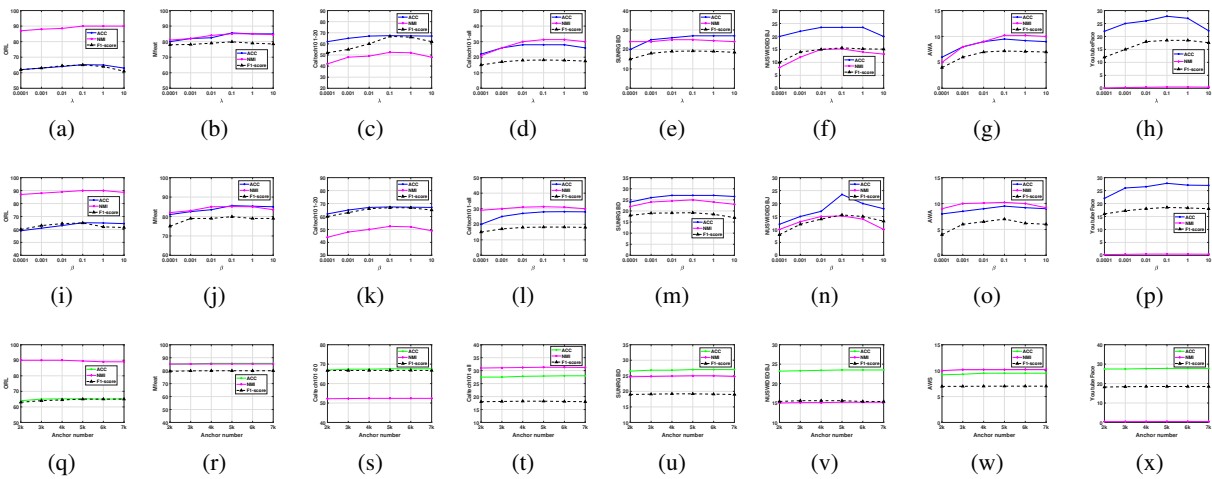

Figure 1. The first and second line are parameter selection of $\lambda$, $\beta$ on eight datasets, respectively. The third line is sensity investigation of anchor number on eight datasets.

Table 1. Clustering results based on ACC (%) on all datasets. "N/A " denotes out of memory.

| Dataset | AMGL | SFMC | BMVC | LMVSC | MSGL | FPMVS | EOMSC-CA | OMSC | Ours |
|---|---|---|---|---|---|---|---|---|---|
| ORL | 64.50±0.01 | 61.40±0.05 | 48.70±0.05 | 58.60±0.02 | 21.00±0.05 | 52.00±0.50 | 62.20±0.05 | 63.80±0.00 | **65.20±0.00** |
| Mfeat | 82.60±0.02 | 75.50±0.20 | 69.30±0.05 | 81.75±0.05 | 75.40±0.02 | 82.20±0.05 | 82.25±0.03 | 84.00±0.05 | **85.60±0.00** |
| Caltech101-20 | 28.70±0.20 | 59.40±0.05 | 16.80±0.05 | 29.00±0.30 | 48.00±0.02 | 66.15±0.10 | 64.10±0.50 | 65.00±0.10 | **67.40±0.00** |
| Caltech101-all | 14.80±0.01 | 17.70±0.05 | 21.20±0.03 | 15.50±0.01 | 14.10±0.02 | 27.50±0.05 | 22.30±0.03 | 24.00±0.00 | **28.00±0.50** |
| SUNRGBD | 9.80±0.01 | 11.30±0.05 | 16.70±0.01 | 18.00±0.05 | 13.00±0.01 | 23.40±0.05 | 23.70±0.05 | 25.20±0.00 | **27.00±0.00** |
| NUSWIDEOBJ | N/A | 12.20±0.05 | 12.90±0.05 | 14.70±0.05 | 12.00±0.05 | 19.20±0.05 | 19.60±0.05 | 21.00±0.05 | **23.50±0.00** |
| AWA | N/A | 3.92±0.03 | 8.60±0.05 | 7.20±0.03 | 8.00±0.02 | 8.90±0.01 | 8.65±0.05 | 9.00±0.10 | **10.50±0.10** |
| YoutubeFace | N/A | N/A | 8.90±0.05 | 14.00±0.02 | 16.70±0.01 | 23.00±0.03 | 26.45±0.05 | 26.50±0.00 | **27.80±0.00** |

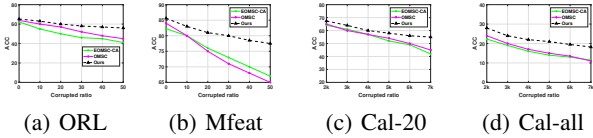

Figure 2. Robustness study of our method on datasets under ACC.

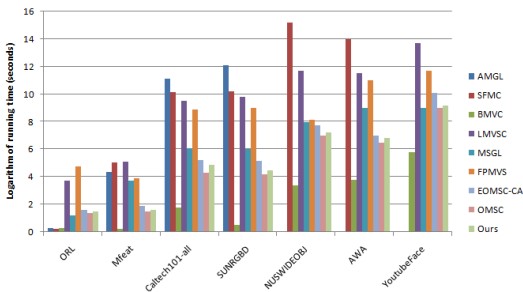

Figure 3. Logarithm of running time on different datasets.

and F1-score. A high value for each of these metrics indicates better clustering performance on the dataset.

We first study how parameters $\beta$ and $\lambda$ influence the final clustering performance. These two parameters are adopted to negotiate the importances of partition term and Frobenious norm term. We illustrate the clustering performance of the proposed method with varying parameters $\lambda$ and $\beta$ in Fig. 1. It is observed that appropriate values for these two parameters are generally beneficial to the clustering results on different datasets. According to Fig. 1, we observe that relatively desired clustering results are achieved when $\beta = 0.1$ and $\lambda = 0.1$ on various datasets. Moreover, the results of the proposed method are generally stable over varying values within the range of parameters $\beta$ and $\lambda$ on different datasets, which shows that RCSC is generally robust to these two parameters.

## 3.2. Experimental results and analyis

In this section, the proposed method is compared with the eight representative methods on several multi-view datasets. To be specific, we report the clustering results with respect to ACC, NMI and F1-score of all multi-view clustering methods in Tables 1-3, respectively. We adopt N/A to indicate that the method is not able to be computationally feasible on the dataset caused by out of memory. Based on the obtained clustering results in Tables 1-3, we can draw some conclusions as follows:

*Table 2.* Clustering results based on NMI (%) on all datasets. "N/A " denotes out of memory.

| Dataset | AMGL | SFMC | BMVC | LMVSC | MSGL | FPMVS | EOMSC-CA | OMSC | Ours |
|---|---|---|---|---|---|---|---|---|---|
| ORL | 87.10±0.07 | 82.70±0.01 | 67.70±0.03 | 78.50±0.03 | 43.70±0.02 | 74.40±0.05 | 88.10±0.02 | 88.50±0.10 | **90.00±0.00** |
| Mfeat | 84.70±0.05 | 84.80±0.10 | 66.05±0.15 | 76.00±0.20 | 76.54±0.05 | 79.40±0.01 | 83.20±0.15 | 84.20±0.10 | **85.32±0.15** |
| Caltech101-20 | 47.50±0.20 | 42.80±0.00 | 16.20±0.03 | 41.20±0.10 | 31.50±0.05 | **63.30±0.05** | 51.10±0.05 | 51.80±0.30 | 52.40±0.00 |
| Caltech101-all | 35.30±0.01 | 26.10±0.03 | **42.50±0.04** | 33.30±0.02 | 26.10±0.02 | 34.10±0.05 | 24.65±0.05 | 30.00±0.00 | 31.30±0.00 |
| SUNRGBD | 18.50±0.10 | 2.30±0.05 | 19.50±0.05 | 24.50±0.05 | 9.30±0.05 | 24.10±0.05 | 22.50±0.01 | 24.30±0.00 | **25.00±0.10** |
| NUSWIDEOBJ | N/A | 0.96±0.01 | 12.90±0.02 | 12.80±0.05 | 5.70±0.03 | 13.20±0.05 | 13.20±0.15 | 14.00±0.00 | **15.20±0.00** |
| AWA | N/A | 0.30±0.05 | 9.70±0.02 | 8.50±0.05 | 7.90±0.03 | 8.50±0.03 | 9.70±0.03 | 10.00±0.02 | **10.22±0.00** |
| YoutubeFace | N/A | N/A | 5.90±0.05 | **11.80±0.01** | 0.07±0.01 | 2.40±0.01 | 0.32±0.01 | 0.37±0.00 | 0.50±0.00 |

*Table 3.* Clustering results based on F1-score (%) on all datasets. "N/A " denotes out of memory.

| Dataset | AMGL | SFMC | BMVC | LMVSC | MSGL | FPMVS | EOMSC-CA | OMSC | Ours |
|---|---|---|---|---|---|---|---|---|---|
| ORL | 51.20±0.03 | 30.60±0.05 | 30.50±0.04 | 45.90±0.09 | 51.50±0.20 | 38.40±0.15 | 62.10±0.00 | 63.20±0.10 | **65.00±0.00** |
| Mfeat | 79.80±0.05 | 71.10±0.15 | 58.80±0.01 | 72.50±0.02 | 70.10±0.02 | 76.00±0.40 | 77.00±0.01 | 78.20±0.10 | **79.90±0.00** |
| Caltech101-20 | 21.80±0.05 | 31.60±0.02 | 11.40±0.20 | 25.60±0.50 | 41.80±0.05 | 66.00±0.05 | 64.70±0.20 | 65.00±0.10 | **66.79±0.20** |
| Caltech101-all | 4.05±0.10 | 4.65±0.10 | 18.00±0.05 | 10.50±0.05 | 8.60±0.04 | 17.90±0.03 | 10.80±0.03 | 15.00±0.00 | **18.20±0.15** |
| SUNRGBD | 6.40±0.40 | 12.10±0.00 | 10.20±0.01 | 11.60±0.20 | 9.50±0.15 | 16.00±0.05 | 15.30±0.05 | 17.00±0.00 | **19.20±0.00** |
| NUSWIDEOBJ | N/A | 11.50±0.01 | 8.80±0.02 | 9.30±0.05 | 8.50±0.05 | 13.50±0.07 | 13.60±0.05 | 14.50±0.00 | **15.60±0.10** |
| AWA | N/A | 4.60±0.03 | 5.59±0.02 | 3.60±0.05 | 4.20±0.01 | 6.20±0.05 | 5.90±0.05 | 6.20±0.00 | **7.00±0.20** |
| YoutubeFace | N/A | N/A | 5.80±0.02 | 8.30±0.01 | 15.00±0.10 | 14.00±0.05 | 16.40±0.01 | 17.10±0.00 | **18.50±0.00** |

. For most datasets, the proposed method achieves more desired performance under different metrics and still behaves well on multi-view datasets with relatively large scale. For example, the clustering performance gain of the proposed method is about 2.32% higher than MSGL in terms of NMI on AWA.

. Methods based on anchor tend to generate better performance under three metrics in most cases on multi-view datasets with large scales compared with general graph-based methods, which demonstrates that buiding the graph based on the anchor is helpful to handle the multi-view datasets with large scales.

. Our method produces consitently better results than other methods based on the anchor for most of the multi-view datasets, which validates the necessarity of ensuring the robustness of learned consensus anchors for efficient multi-view subspace clustering and exploring the correlation among the learned consensus anchors with the guidance of view-specific projection in the manner of encouraging the grouping effect and grouping highly correlated anchors together.

### 3.3. Sensitivity Investigation and Robustness Study

We investigate how the total number of anchors impacts the clustering results in this part. For simplicity, we fix the shared dimension and conduct the sensity analysis for the number of anchors on several datasets in terms of different metrics. According to Fig. 1, we find that the proposed method is not significantly influenced by the number of anchors and the clustering results with different number of anchors are relatively stable.

We also study the robustness of the proposed method on

different datasets. To be specific, we randomly select half of the multi-view dataset to be corrupted with white Gaussian noise. This type of noise is added to the selected data point $x_i^p$ via $\tilde{x}_i^p = x_i^p + pj$, where $\tilde{x}_i^p \in [0, 255]$, $p$ denotes the corrupted ratio and $j$ is the noise satisfying the standard Gaussian distribution. According to Fig. 2, we can observe that the proposed method is robust on different datasets compared with other methods and performs better on these datasets, which can be explained by the fact that the ensuring the robustness of learned consensus anchors in the affine subspace for efficient multi-view subspace clustering is helpful in achieving satisfied performance.

### 3.4. Running Time

We report the execution times of the compared methods and ours on different datasets. Note that Caltech101-20 and Caltech101-all are two versions of Caltech101 dataset and we just list the running time of Caltech101-all for simplicity. As shown in Fig. 3, it is observed that the proposed method has shown comparable logarithm of running time cost to the existing efficient methods on most of the multi-view datasets, i.e., MSGL. Thus, our method can obtain advantageous clustering results on various datasets while maintaining relatively competitive efficiency. It can be explained by the fact that jointly modeling the robust consensus anchors and the common cluster structure in a unified framework is crucial to guide the efficiency for multi-view clustering. The extra clustering algorithm is not needed to obtain the final results, i.e., spectral clustering.

## 4. Conclusion

We propose a novel RCSC in this work. We theoretically demonstrate that a block-diagonal anchor graph is obtained

if the objective function satisfies certain conditions. As a special case, we give a model based on Frobenius norm, non-negative and affine constraints in consensus anchors learning, which guarantees the robustness of learned consensus anchors for efficient multi-view clustering and investigates the specific local distribution of cluster in the affine subspace. Extensive experiments verify the effectiveness and efficiency of the proposed method on different multi-view datasets under three metrics.

## Acknowledgments

The work was partially supported by Eastern Talent Plan Leading Project under Grant BJKJ2024011 and National Natural Science Foundation of China (62402303).

## Impact Statement

This paper presents work whose goal is to advance the field of Machine Learning. There are many potential societal consequences of our work, none which we feel must be specifically highlighted here.

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
