# OpenReview forum: "Robust Consensus Anchor Learning for Efficient Multi-view Subspace Clustering"
_ICML.cc/2025/Conference — ICML 2025 poster_

### Official Review · Reviewer_MKbn · 2025-03-08

**Overall Recommendation:** 4

**Summary:**

This paper proposes a novel Robust Consensus anchor learning for efficient multi-view Subspace Clustering (RCSC). The authors first theoretically demonstrate that an anchor graph with block-diagonal structure can be achieved if the objective function satisfies certain conditions. The authors impose the orthogonal constraints on the actual bases and constrain a factor matrix to be the cluster indicator matrix built on the rigorous clustering interpretation. Extensive experiments on different multi-view datasets validate the effectiveness and efficiency of RCSC, especially on the datasets with large scales.

**Claims And Evidence:**

Yes,

**Essential References Not Discussed:**

No.

**Ethical Review Concerns:**

N/A.

**Experimental Designs Or Analyses:**

Yes. The comparison of experimental methods consists of methods from the recent years, which increases the credibility of the experimental results.

**Methods And Evaluation Criteria:**

Yes.

**Other Comments Or Suggestions:**

No.

**Other Strengths And Weaknesses:**

Strengths:
This paper imposes the orthogonal constraints on the actual bases and constrain a factor matrix to be the cluster indicator matrix built on the rigorous clustering interpretation. Extensive experiments on different multi-view datasets validate the effectiveness and efficiency of RCSC, especially on the datasets with large scales.


Weaknesses:
1. The authors adopt $ A\in R^{d\times l} $ to represent the unified anchors, $ l $ and $ d $ are the number of anchors and shared dimension across views, respectively. $ U^{p}A\in R^{d_{p}\times l} $ represents the basis matrix. The authors then theoretically demonstrate that a block-diagonal anchor graph can be achieved if the corresponding objective function satisfies certain conditions based on the independent subspace assumption, which is shown as Theorem 1 in the following. However, the authors do not give the explanation why $ U^{p}A\in R^{d_{p}\times l} $ can represent the basis matrix and the related analysis can be given here.
2. The authors state that the data are usually contaminated with the possible noise in real applications and then adopt the Frobenius norm for penalizing the noise based on affine and non-negative constraint in Eq. (5). Here, the authors need to explain what kind of noise is dealt with by the Frobenius norm.
3. The authors state that the objective function monotonically decreases in each iteration until convergence since the convex property for each sub-problem and then list the procedure of RCSC in Algorithm 1. The authors are expected to give some explanation for the procedure of RCSC in Algorithm 1.
4. The authors tune the number of the proposed method in the range of $ [2k,3k,\cdots,7k] $, where $ k $ denotes the total number of clusters in dataset. The authors are expected to give some analysis why choose $ [2k,3k,\cdots,7k] $ as the range instead of the others in this part.

**Questions For Authors:**

1. The authors use $ (U^{p}A)^{t} $ as the basis matrix lying in the $ t $-th affine subspace. Here, the dimension of $ (U^{p}A)^{t} $ is expected to be given to better show the reason why it can be adopted as the basis matrix lying in the $ t $-th affine subspace.
2. The authors do not give the notations table throughout the paper and the physical meaning of each variable is not very clear. Then the authors need to give the notation table in this paper.

**Relation To Broader Scientific Literature:**

Compared to the previous studies, the highlights of this paper fully explore the correlation among the learned consensus anchors with the guidance of view-specific projection, which encourages the grouping effect and tends to group highly correlated anchors together.

**Theoretical Claims:**

Yes.  There have proofs for theoretical claims in the paper and they are checked the correctness.

---

> ### Author Rebuttal · Authors · 2025-03-28
>
> Q1: The explanation why UpA∈Rdp×1 can represent the basis matrix and the related analysis can be given here.
>
> A1: Good question! As reviewer mentioned, we adopt A∈Rd×l to represent the unified anchors, l and d are the number of anchors and shared dimension across views, respectively. UpA∈Rdp×l represents the basis matrix. We then theoretically demonstrate that a block-diagonal anchor graph can be achieved if the corresponding objective function satisfies certain conditions based on the independent subspace assumption, which is shown as Theorem 1 in the following. However, we do not give the explanation why UpA∈Rdp×l can represent the basis matrix and the related analysis can be given here. The reason why we use UpA∈Rdp×l to represent the basis matrix is that the dimension of UpA corresponds to the dimension of the basis matrix for the p-th view. We will add this explanation in the camera-ready version.
>
> Q2: What kind of noise is dealt with by the Frobenius norm in this work?
>
> A2: Thanks for the comment! The data are usually contaminated with the possible noise in real applications and we then adopt the Frobenius norm for penalizing the noise based on affine and non-negative constraint in Eq. (5). The Frobenius norm is adopted to deal with the Gaussian noise in this work.
>
> Q3: The authors should give some explanation for the procedure of RCSC in Algorithm 1.
>
> A3: Good question! As reviewer pointed, we state that the objective function monotonically decreases in each iteration until convergence since the convex property for each sub-problem and the procedure of RCSC is listed in Algorithm 1. We are expected to give some explanation for the procedure of RCSC in Algorithm 1. Given multi-view dataset, parameter \lambda , \beta ,number of clusters k ,we first initialize A , U^{p} , {\alpha{p}}_{p=1}^{v} , S , F , G and achieve the cluster assignment F after repeating the six updating steps in Algorithm 1. We will add these details in the camera-ready version.
>
> Q4: The reason why the range of [2k,3k,…,7k] is adopted in the experiment should be given.
>
> A4: Thanks for the comment! We tune the number of the proposed method in the range of [2k,3k,…,7k], where k denotes the total number of clusters in the dataset. The reason why we choose [2k,3k,…,7k] as the range instead of the others is that such range represents the representative magnitudes in the experiment.
>
> Q5: The dimension of (UpA)t can be given to better show the reason why it can be adopted as the basis matrix lying in the t-th affine subspace.
>
> A5: Good question! As reviewer pointed, we should give the dimension of (UpA)t to better show the reason why it can be adopted as the basis matrix lying in the t-th affine subspace. The dimension of (UpA)t is dp×l, which corresponds to the dimension of the basis matrix lying in the t-th affine subspace. We will add this explanation in the camera-ready version.
>
> Q6: The authors need to the give the notation table in the paper.
>
> A6: Thanks for the comment! It is needed to give the notation table throughout the paper to make the physical meaning of each variable very clear. We will give the notation table for the camera-ready version and omit the details here for simplicity.

---

### Official Review · Reviewer_1Fds · 2025-03-08

**Overall Recommendation:** 3

**Summary:**

To improve the scalability of the multi-view subspace clustering to large-scale data, this paper proposes Robust Consensus anchors learning for efficient multi-view Subspace Clustering (RCSC), which joints the robust anchor learning, anchor graph construction, and partition into a unified framework. This paper theoretically demonstrates that a optimal anchor graph with block-diagonal structure is obtained if the objective function satisfies certain conditions. The RCSC is simple and effective in terms of performance.

**Claims And Evidence:**

Yes.

**Essential References Not Discussed:**

No.

**Experimental Designs Or Analyses:**

Yes. The authors perform extensive experiments on different benchmark datasets to demonstrate the effectiveness and efficiency of the proposed method.

**Methods And Evaluation Criteria:**

Yes.

**Other Comments Or Suggestions:**

The Abstract is too long to read.

**Other Strengths And Weaknesses:**

Strength:

To improve the scalability of the multi-view subspace clustering to large-scale data, this paper proposes Robust Consensus anchors learning for efficient multi-view Subspace Clustering (RCSC), which joints the robust anchor learning, anchor graph construction, and partition into a unified framework.

Weakness:

1. Authors claim that "Besides, the robust consensus anchors and the common cluster structure shared by different views are not able to be simultaneously learned in a unified framework". However, some methods can already achieve this, for example, FPMVS-CAG(Fast Parameter-free Multi-view Subspace Clustering with Consensus Anchor Guidance) and SMVSC(Scalable Multi-view Subspace Clustering with Unified Anchors). It is crucial to analyze and explain the connection and differences with the above methods, especially the SMVSC.

2. In Lemma 1, the symbol ||A||_* is the nuclear norm? What is the function of Lemma 1? It seems that there is no nuclear norm in the objective function.

3. Some symbol definitions are unclear, such as X_i^P, A_i.

4. The data used in the experiment is unclear, such as the scale of the YoutubeFace.

5. The experimental section lacks the latest and SOTA methods for comparison.

6. In Figure 4, the horizontal axis is Corrupted ratio, but why is 2k, 3k, 4k, 5k,6k, 7k in (c) and (d)?

**Questions For Authors:**

1. The experimental section lacks the latest and SOTA methods for comparison.
2. In Figure 4, the horizontal axis is Corrupted ratio, but why is 2k, 3k, 4k, 5k,6k, 7k in (c) and (d)?

**Relation To Broader Scientific Literature:**

This paper theoretically demonstrates that an optimal anchor graph with block-diagonal structure is obtained if the objective function satisfies certain conditions. The RCSC is simple and effective in terms of performance.

**Theoretical Claims:**

Yes. The authors give proofs for theoretical claims in this work and I have carefully checked them.

---

> ### Author Rebuttal · Authors · 2025-03-28
>
> Q1: It is important to emphasize the connection and differences between the following works: i.e., FPMVS-CAG and SMVSC, especially SMVSC.
>
> A1: Good question! FPMVS-CAG jointly performs anchor selection and subspace graph construction into a framework. Then the two processes can be negotiated with each other to improve the clustering performance. SMVSC integrates anchor learning and graph construction into a unified optimization process. A more discriminative clustering structure can be achieved in this manner. The connection between the above two works and ours is that anchor learning and subspace graph construction are simultaneously conducted in a unified framework. The differences between these two works and ours, especially SMVSC, are that we mainly focus on the robust consensus anchor learning as the highlighted in the title. We first theoretically demonstrate that an anchor graph with block-diagonal structure can be achieved if the objective function satisfies certain conditions. As a special case, we give a model based on Frobenius norm, non-negative and affine constraints in consensus anchors learning, which guarantees the robustness of learned consensus anchors for efficient multi-view clustering and investigates the specific local distribution of cluster in the affine subspace. We will add these analysis in the camera-ready version.
>
> Q2: In Lemma 1, the symbol ||A||_* is the nuclear norm? What is the function of Lemma 1? It seems that there is no nuclear norm in the objective function.
>
> A2: Good question! The symbol ||A|| _* is the nuclear norm in Lemma 1. Lemma 1 mainly gives an inequation to support the conclusion with ||S|| _ *>=||W|| _ * in Theorem 3. Then W is optimal and owns the block diagonal structure. Both left and right of inequation in Lemma1 are nuclear norms and we will further highlight it for the camera-ready version.
>
> Q3: Some symbol definitions are unclear, such as X_i^p, A_i.
>
> A3: Thanks for the comment! As reviewer mentioned, some symbol definitions are unclear, i.e., X_i^p denotes the data points belonging to the i-th cluster for p-th view. A_i is the anchors belonging to the i-th cluster. We will add these explanations and check the whole paper to avoid the similar issues.
>
> Q4: The data used in the experiment is unclear, i.e., the scale of YoutubeFace.
>
> A4: Good question! It is needed to clearly show the data in the experiment as reviewer pointed. YoutubeFace has total 101499 instances generated from YoutTube and we will add it for the camera-ready version.
>
> Q5: The experimental section lacks the latest and SOTA methods for comparison.
>
> A5: Thanks for the comment! We have added two latest and SOTA methods for comparison in the experimental section, i.e., OMVCDR [a] and RCAGL [b].
>
> [a] One-Step Multi-View Clustering With Diverse Representation, 2024
>
> [b] Robust and Consistent Anchor Graph Learning for Multi-View Clustering, 2024
>
> The clustering results of OMVCDR based on ACC for all datasets are 64.00±0.00, 84.20±0.00, 65.75±0.00, 24.32±0.00, 25.50±0.10, 21.26±0.00, 9.18±0.00, and 26.79±0.00.
>
> The clustering results of RCAGL based on ACC for all datasets are 64.13±0.05, 84.17±0.00, 65.28±0.10, 25.00±0.00, 25.35±0.00, 22.00±0.05, 9.25±0.00, and 26.92±0.05.
>
> The clustering results of OMVCDR based on NMI for all datasets are 88.72±0.00, 84.35±0.00, 51.95±0.00, 30.27±0.00, 24.27±0.00, 14.18±0.00, 9.98±0.00, and 0.34±0.00.
>
> The clustering results of RCAGL based on NMI for all datasets are 88.90±0.00, 84.69±0.05, 51.86±0.00, 30.50±0.05, 24.68±0.00, 14.40±0.00, 10.10±0.00, and 0.35±0.02.
>
> The clustering results of OMVCDR based on F1-score for all datasets are 63.15±0.00, 78.30±0.00, 65.20±0.00, 15.80±0.00, 17.59±0.00, 14.82±0.00, 6.42±0.00, and 17.35±0.00.
>
> The clustering results of RCAGL based on F1-score for all datasets are 63.80±0.00, 78.85±0.00, 65.14±0.10, 16.37±0.00, 18.10±0.00, 15.00±0.00, 6.57±0.00, and 17.70±0.00.
>
> Q6: In Figure 4, the horizontal axis is Corrupted ratio, but why is 2k, 3k, 4k, 5k, 6k, 7k in (c) and (d)?
>
> A6: Thanks very much for carefully reading our paper! The horizontal axis should be Corrupted ratio in Figure 4 and we wrongly wrote is as 2k, 3k, 4k, 5k, 6k, 7k in (c) and (d). We will correct this typo and check the whole paper to avoid the similar issues.
>
> Q7: Reduce the length of the Abstract.
>
> A7: Thanks for the comment! It is needed to streamline and differentiate some details in the abstract, which is able to improve the overall readability and appeal of the manuscript. We will remove the related details regarding the significant overlapped parts with the contribution summary section for the camera-ready version.

---

> > ### Comment · Reviewer_1Fds · 2025-04-03
> >
> > Thanks for your response, my questions are partially solved, and if the author can complete the revision in the final version, I will consider raising the score.

---

### Official Review · Reviewer_y2ao · 2025-03-10

**Overall Recommendation:** 4

**Summary:**

This paper proposes Robust Consensus anchors learning for efficient multi-view Subspace Clustering (RCSC). The authors first show that if the data are sufficiently sampled from independent subspaces, and the objective function meets some conditions, the achieved anchor graph has the block-diagonal structure. As a special case, the authors provide a model based on Frobenius norm, non-negative and affine constraints in consensus anchors learning, which guarantees the robustness of learned consensus anchors for efficient multi-view clustering and investigates the specific local distribution of cluster in the affine subspace. While it is simple, we theoretically give the geometric analysis regarding the formulated RCSC. The union of these three constraints is able to restrict how each data point is described in the affine subspace with specific local distribution of cluster for guaranting the robustness of learned consensus anchors.

**Claims And Evidence:**

Yes.

**Essential References Not Discussed:**

No.

**Experimental Designs Or Analyses:**

Yes. The adopted methods for comparison include the works from the recent years, increasing the credibility of the final results.

**Methods And Evaluation Criteria:**

Yes.

**Other Comments Or Suggestions:**

No.

**Other Strengths And Weaknesses:**

Strength: It is able to ensure the mutual enhancement for these procedures and helps lead to more discriminative consensus anchors as well as the cluster indicator. We then adopt an alternative optimization strategy for solving the formulated problem.


Weakness:
1. The authors adopt the orthogonal and nonnegative factorization to directly assign clusters to the data for integrating the partition into the unified framework. The authors should explain why the partition can be integrated into the unified framework based on the orthogonal and nonnegative factorization.

2. The authors report the clustering results with respect to ACC, NMI and F1-score of all multi-view clustering methods in Tables 1-3, respectively. Then they adopt N/A to indicate that the method is not able to be computationally feasible on the dataset caused by out of memory. The authors are expected to highlight the second best clustering performance in Tables 1-3 to make the performance gains of the proposed method more obvious.

3. The authors fix the shared dimension and conduct the sensity analysis for the number of anchors on several datasets in terms of different metrics. According to Fig. 3, The authors find that the proposed method is not significantly influenced by the number of anchors and the clustering results with different number of anchors are relatively stable. In this part, the authors should give more detailed analysis regarding how the total number of anchors impacts the clustering results in this part.

**Questions For Authors:**

1. The authors report the execution times of the compared methods and theirs on different datasets. As shown in Fig. 5, it is observed that the proposed method has shown comparable logarithm of running time cost to the existing efficient methods on most of the multi-view datasets, i.e., MSGL. However, the authors do not give the memory value of the used device, which is important in running time analysis part.

2. In this reference part, some names of publication are termed for short, i.e., Artif. Intell., however, some names of publications are fully termed, i.e., IEEE Conference on Computer Vision and Pattern Recognition. I think the authors should carefully check this issue to improve the whole presentation of this paper.

**Relation To Broader Scientific Literature:**

This paper first shows that if the data are sufficiently sampled from independent subspaces, and the objective function meets some conditions, the achieved anchor graph has the block-diagonal structure. As a special case, the authors provide a model based on Frobenius norm, non-negative and affine constraints in consensus anchors learning, which guarantees the robustness of learned consensus anchors for efficient multi-view clustering and investigates the specific local distribution of cluster in the affine subspace.

**Theoretical Claims:**

Yes. There are proofs for theoretical claims in the paper and I have checked them.

---

> ### Author Rebuttal · Authors · 2025-03-28
>
> Q1: The reason why the partition can be integrated into the unified framework.
>
> A1: Thanks for the comment! To integrate the partition into the unified framework, we adopt the orthogonal and nonnegative factorization to directly assign clusters to the data. The reason why we integrate the partition into the unified framework is that extra post-processing steps are not needed in recovering cluster structures based on the factor matrix. Specifically, we impose the orthogonal constraint on the actual bases. We will add such explanation for the camera-ready version.
>
>
> Q2: The second best results in Tables 1-3 should be highlighted.
>
> A2: Good question! It is needed to highlight the second best clustering performance in Tables 1-3 to make the performance gains of the proposed method more obvious. We will highlight the second best clustering performance in Tables 1-3 of the experiment for the camera-ready version.
>
>
> Q3: The detailed analysis regarding how the anchor number impacts the results.
>
> A3: Thanks for the comment! According to Fig. 1, we find that the proposed method is not significantly influenced by the number of anchors and the clustering results with different number of anchors are relatively stable. Besides, larger number of anchors tends to achieve more desired clustering performance and not too large anchor number will help produce relatively satisfied clustering performance. We will add the above descriptions for the camera-ready version.
>
>
> Q4: The memory value of the used device should be given in running time analysis.
>
> A4: Good question! It is needed to give the memory value of the used device in running time analysis. The memory of the adopted device in running time analysis for the experiment is 8G and we will add it for the camera-ready version.
>
>
> Q5: The consistency for the formants of reference should be ensured.
>
> A5: Thanks for the comment! We will correct the issue mentioned by the reviewer for the inconsistency issue for the formants of reference and correct the whole reference part to avoid the similar issues for the camera-ready version.

---

### Official Review · Reviewer_Y9MR · 2025-03-12

**Overall Recommendation:** 4

**Summary:**

This study proposes a novel method named RCSC, which aims to improve the clustering effectiveness on multi-view datasets by jointly addressing anchor graph construction, partitioning, and robust anchor learning. A key finding is that when data are adequately sampled from independent subspaces and the objective function satisfies certain conditions, the resulting anchor graph exhibits a block diagonal structure. This structure facilitates the revelation of specific local distributions of clusters within affine subspaces. Additionally, the research underscores the importance of robustness in consensus anchors. This emphasis on robustness ensures that the consensus anchors can reliably represent the underlying data structures across different views, thereby enhancing the overall clustering quality.

**Claims And Evidence:**

When proving Theorem 3, it is mentioned at the end that "we find that the proportions of vertices are far more less than edge and inside basis in practice." Although this observation is pointed out, there is no in-depth exploration of the reasons or mechanisms behind it. Providing some tentative theoretical explanations, even if preliminary, would also enhance the persuasiveness of the argument.

**Essential References Not Discussed:**

No.

**Experimental Designs Or Analyses:**

Yes, I have reviewed the soundness and validity of the experimental designs and analyses conducted in this study.

**Methods And Evaluation Criteria:**

The proposed methods and evaluation criteria are indeed well-suited for addressing the problem.

**Other Comments Or Suggestions:**

See weaknesses above.

**Other Strengths And Weaknesses:**

Strengths:
1. The paper demonstrates a clear writing approach, providing detailed and well-organized formula derivations that guide the reader from the problem definition to the gradual introduction of various constraints.

2.The paper develops a novel approach for multi-view subspace clustering.

Weaknesses:
1.The abstract is somewhat verbose and lacks conciseness, with significant overlap with the contributions summary section. It is recommended that the authors streamline and differentiate these sections to improve the overall readability and appeal of the manuscript.

2.To further enhance the readers' understanding, it is suggested to incorporate more schematic diagrams or graphical illustrations at appropriate places within the text. This will help in providing a more intuitive presentation of complex concepts and relationships.

3. While the extensive experimentation conducted is noteworthy, the sheer volume of experimental data has resulted in many figures being too small to examine the results thoroughly. The authors should consider employing more effective layout strategies or alternative solutions to optimize the presentation of these figures, ensuring that all critical data are clearly visible.

**Questions For Authors:**

See weaknesses above.

**Relation To Broader Scientific Literature:**

The paper emphasizes the importance of the robustness of consensus anchors and reducing computational complexity. These improvements are crucial for dealing with large-scale datasets, aligning with the research objectives in the fields of machine learning and data mining - developing algorithms that can operate efficiently while maintaining high accuracy. Moreover, the RCSC method promotes a grouping effect, which means combining highly correlated consensus anchors to generate more distinctive cluster indicators. This approach is significant for exploring the deeper structures within data.

**Theoretical Claims:**

Yes, I have checked the correctness of any proofs for theoretical claims, such as Theorem 1-3, and verified the correctness of all formulas. Specifically, a more detailed description of how Eq.(16) was derived can be provided

---

> ### Author Rebuttal · Authors · 2025-03-28
>
> Q1: Streamline and differentiate some details in the abstract.
>
> A1: Thanks for the comment! It is needed to streamline and differentiate some details in the abstract, which is able to improve the overall readability and appeal of the manuscript. We will remove the related details regarding the significant overlapped parts with the contribution summary section for the camera-ready version.
>
>
> Q2: Incorporate more schematic diagrams or graphical illustrations at appropriate places within the text.
>
> A2: Good question! It is needed to incorporate more schematic diagrams or graphical illustrations at appropriate places within the text for further enhancing the readers’ understanding. It will also help in providing a more intuitive presentation of complex concepts and relationships. For example, we will add the specific names of the datasets adopted for parameter selection and sensity investigation of anchor number in the title of Figure 1.
>
>
> Q3: Optimize the presentation of figures in the paper.
>
> A3: Thanks for the comment! We will consider employing more effective layout strategies or alternative solutions as reviewer mentioned to optimize the presentation of the figures to ensure that all critical data are clear visible, which is able to avoid that many figures are too small to examine the results thoroughly. For example, we will use more lines and increase the size of all figures in Figure 1 for parameter selection and sensity investigation of anchor number.

---

> > ### Comment · Reviewer_Y9MR · 2025-04-04
> >
> > My comments have been partially addressed in the rebuttal. I decide to raise my rating to accept.

---

### Decision · Program_Chairs · 2025-05-01

**Decision:**

Accept (poster)

**Comment:**

To address anchor graph construction, partitioning, and robust anchor learning, this paper proposes the RCSC method to improve the clustering effectiveness on multi-view datasets. All reviewers give positive scores. This paper has clear writing. The experiment is reasonable.